# Vaping-Dependent Pulmonary Inflammation Is Ca^2+^ Mediated and Potentially Sex Specific

**DOI:** 10.3390/ijms25031785

**Published:** 2024-02-01

**Authors:** Jeffrey G. Shipman, Rob U. Onyenwoke, Vijay Sivaraman

**Affiliations:** 1Department of Biological & Biomedical Sciences, North Carolina Central University, Durham, NC 27707, USA; jshipma7@eagles.nccu.edu (J.G.S.); ronyenwo@nccu.edu (R.U.O.); 2Biomanufacturing Research Institute and Technology Enterprise (BRITE), North Carolina Central University, Durham, NC 27707, USA; 3The Julius L. Chambers Biomedical/Biotechnology Research Institute, North Carolina Central University, Durham, NC 27707, USA

**Keywords:** calcium (Ca^2+^), inflammation, pulmonary biology, macrophages, neutrophils, vaping, pathology

## Abstract

Here we use the SCIREQ InExpose system to simulate a biologically relevant vaping model in mice to investigate the role of calcium signaling in vape-dependent pulmonary disease as well as to investigate if there is a gender-based difference of disease. Male and female mice were vaped with JUUL Menthol (3% nicotine) using the SCIREQ InExpose system for 2 weeks. Additionally, 2-APB, a known calcium signaling inhibitor, was administered as a prophylactic for lung disease and damage caused by vaping. After 2 weeks, mice were exposed to lipopolysaccharide (LPS) to mimic a bacterial infection. Post-infection (24 h), mice were sacrificed, and bronchoalveolar lavage fluid (BALF) and lungs were taken. Vaping primed the lungs for worsened disease burden after microbial challenge (LPS) for both males and females, though females presented increased neutrophilia and inflammatory cytokines post-vape compared to males, which was assessed by flow cytometry, and cytokine and histopathological analysis. This increased inflammatory burden was controlled by calcium signaling inhibition, suggesting that calcium dysregulation may play a role in lung injury caused by vaping in a gender-dependent manner.

## 1. Introduction

Respiratory diseases such as acute respiratory distress syndrome (ARDS), asthma, and chronic obstructive pulmonary disease (COPD) represent a significant health burden for morbidity and mortality worldwide [1]. These lung diseases are characterized by lung inflammation, which is mediated by macrophages, neutrophils, and other immune cells that release chemical messengers known as cytokines and chemokines that modulate this inflammation [2,3]. This inflammation is present within one or both lungs and is typically caused by infection with viruses, bacteria, or fungal pathogens with prognosis known to worsen due to exposure to airway pollutants [4,5,6]. The societal intake of tobacco and the use of various smoking products such as electronic cigarettes (E-cigs, also known as “vaping”) has been linked to worsened lung health and increasing susceptibility to infections, leading to these pulmonary diseases [7,8].

Vaping has become more prevalent in countries worldwide [9,10] and has been initially described as a safer alternative to cigarette smoking [11]. Despite this initial assessment, little is known about whether long-term vaping would have similar effects as cigarette smoke. In 2023, approximately 2.1 million youths, including middle-school- and high-school-aged students, vaped [12]. Vaping studies have also described differences in use among genders. In 2021, it was demonstrated that more men (5.1%) compared to women (4.6%) vaped, with the highest use being among adults 18–24 years of age [13]. Differences in lung diseases based on gender have also been observed, with worsened disease outcomes observed for women [14]. These data suggest that vaping may affect men and women differently [14]. Orzabal and company have described the adverse side effects of chronic vaping, which is known to lead to vascular damage and growth deficits and further illustrates a need for the regulation of vaping for overall safety [15]. Alongside the need for regulation, other research has demonstrated the fact that vaping is associated with acute lung injury and pulmonary inflammation [15,16]. These data underline a need to better characterize the effects of long-term/chronic vaping exposure. Vaping may also be responsible for exposing humans to reactive oxygen species (ROS) similar to those produced from tobacco exposure [17]. ROS are present in cigarette smoke and can lead to lung inflammation. These ROS are a risk factor for the development of COPD and ARDS.

Calcium (Ca^2+^) signaling dysregulation has been shown to play a pivotal role in inflammation and disease burden [6]. However, little is known about the mechanistic action of Ca^2+^ signaling dysregulation and vaping. Ca^2+^ is an important secondary messenger in many cellular processes. In order to maintain Ca^2+^ homeostasis, cells utilize a complex system of channels, pumps, transporters, etc. to mediate intracellular Ca^2+^ levels [18]. The components of Ca^2+^ signaling can be divided into two major groups based on their mediating function: influx/uptake and efflux/release. Ca^2+^ influx components are responsible for the uptake of extracellular Ca^2+^ into the cell while Ca^2+^ release components are responsible for the release of intracellular Ca^2+^ stores. The major Ca^2+^ channels are transient receptor potential channels (TRP), store-operated calcium entry (SOCE) channels, and voltage-gated calcium (VGC) channels. Previous research on tobacco exposure has demonstrated that dysregulation in Ca^2+^ signaling may be responsible for the adverse effects produced by smoking tobacco, with both increases [19] and decreases [20] in intracellular Ca^2+^ concentrations being correlated with cell dysfunction and inflammation. E-cig use, which may have similar health implications as tobacco use, has garnered attention recently in studying its effects on pulmonary inflammation and Ca^2+^ [6].

Here, we use the SCIREQ InExpose system to model the vaping of mice through nose-only administration. This system offers a biologically relevant vaping exposure model in mice compared to our and other previously outlined intranasal models [21,22,23]. We aimed to characterize the effect of JUUL Menthol vape exposure on C57-BL/6 mice of both genders through this more biologically relevant means of vaping. Additionally, we evaluated the role for Ca^2+^ signaling in our model of pulmonary disease exacerbation. The objective of this research was to investigate the role of vaping in inflammation and other cellular responses and to determine if this response can be mitigated by blocking calcium signaling. Our central hypothesis is that prolonged vaping leads to an exacerbation of lung injury and that blocking specific Ca^2+^ signaling components, which are a primary force underlying lung inflammation, may be therapeutic. Importantly, this may present an effective target for therapeutic involvement such that this lung injury can be diminished.

## 2. Results

### 2.1. Establishing an Acute Vaping Mouse Model Using the SCIREQ InExpose System

A challenge with investigating the toxicity and biological effects of vaping resides with the exposure model itself. E-liquids must be quickly heated to a vaporization temperature (approximately 250–300 °C) and drawn in by the user into the pulmonary tract. Prior exposure models used either vape condensate dropwise intranasal [21,23] or intratracheal exposure or whole body vapor exposure [24], which are riddled with complexities that defray their biological relevance. To establish a more biologically relevant vaping model, we utilized the SCIREQ InExpose compact inhalation system/“vape tower” to simulate airway vaping in a nose-only exposure mouse model (Figure 1B). Previous research has shown that an inhaled e-liquid aerosol model offers a more accurate model for cytotoxicity assays as opposed to a condensed e-liquid model [25,26], while not exposing extraneous mucosal sites or skin/fur, which would allow the e-liquid to be orally ingested. During our initial exposure, the mice were vaped for 4 weeks (Figure 1A). Afterwards, the mice were humanely sacrificed, and lungs were collected. The lung wet/dry ratio (a well-described clinical measure of pneumonia/pulmonary inflammation) demonstrated significant changes when comparing the vape group to the mock group (Figure 1C). When investigating total protein using BCA, a significant difference between the mock and vape was not demonstrated (Figure 1C). These data suggest that even after only 4 weeks of vaping alone, slight but significant negative effects on lung health are clearly observable as lungs exposed to vaping had an increase in weight due to fluid and/or inflammatory cells, which are hallmarks of pulmonary exacerbation [27].

### 2.2. An Acute Vaping In Vivo Model Illustrates Vaping Exacerbates Lung Cellularity and Inflammation in the Presence of Infection

To evaluate the effects of vaping on pulmonary disease, lipopolysaccharide (LPS) was used as a surrogate of infection due to its presence within the cell wall of Gram-negative bacteria such as *Klebsiella pneumoniae*, which is a bacterium commonly associated with pulmonary inflammation [23]. Additionally, we wanted to investigate whether the effects of vaping were visible in a shortened time course compared to our prior 4-week exposure. Using our SCIREQ InExpose system, we were able to vape mice for two weeks and then expose them to LPS intranasally. Post-infection animals (24 h) were sacked, and BALF and lungs were collected (Figure 2A). Flow cytometry was performed upon BAL cells to investigate levels of immune cell populations present in the lungs post-LPS exposure (Flow cytometry antibodies listed in Table 1). From the flow cytometry analysis, a significant increase in neutrophil migration into the alveolar space was observed in the vape + LPS mice when compared to the vape alone mouse group. However, when observing the activation of these neutrophils, the increase was not significant (Figure 2B). The inflammatory cytokines tumor necrosis factor alpha (TNF-α) and Interlukin-6 (IL-6) were then investigated using clarified BALF. When comparing the vape alone and vape + LPS groups, both cytokine levels were significantly increased, suggesting a worsened inflammatory state within the vaping-primed lungs (Figure 2C). In IL-6 and TNF-α, no other significance differences were observed. H&E staining of lung sections demonstrated alveolar wall thickening and increased neutrophilia in the vape alone and vape + LPS groups when compared to the mock group (Figure 2D). Our lung injury score demonstrated a significant increase in damage in vape alone groups compared to mock groups. A significant increase was also observed in the vape + LPS group compared to vaping alone. These data suggest that vaping primed lungs have worsened disease burden when infected, which is in line with previous research performed by our lab [21].

### 2.3. Treatment with a Ca^2+^ Inhibitor Mitigates the Alveolar Damage Caused by Vaping and/or Infection

Previous research in our lab has shown Ca^2+^ dysregulation to potentially play a role in vaping-dependent pulmonary disease in vitro and in vivo [23,28]. To evaluate whether Ca^2+^ dysregulation has similar effects upon vaping-dependent disease and inflammation within our new exposure model system, we used the well-described inositol triphosphate (IP_3_) receptor antagonist 2-APB to block endoplasmic reticulum (ER) resident Ca^2+^ release, which is a previously described potential mechanism of vaping toxicity [29]. The SCIREQ InExpose system was again utilized, and a group of mice received 2-APB intranasally (10 µL of a 500 µM stock) each day in addition to being vaped (Figure 3A). On day 16, mice were humanly euthanized, and cells isolated from BALF were analyzed by flow cytometry (as described above). We observed a significant increase in neutrophil migration in the vape + LPS group compared to the LPS alone and vape alone treatment groups, which was similar to the previous experiments. However, this neutrophil migration was significantly diminished in the presence of 2-APB (Figure 3B). A significant decrease in macrophage numbers was also observed in the vape + LPS group compared to vaping alone. This decrease in macrophages was then recovered and increased in the presence of 2-APB (Figure 3B). Inflammatory cytokine (TNF-α and IL-6) levels were also investigated as described above, with IL-6 demonstrating a significant increase in the vape + LPS group compared to the vape alone group (Figure 3C). While an increase in TNF-α levels was observed, there was not a significant difference. H&E staining of lung sections demonstrated alveolar wall thickening and increased neutrophilia in the LPS alone, but this was significantly increased in the vape + LPS group (as seen previously). Vaping resulted in a significant increase in lung injury score when compared to the mock (Figure 3D). Alveolar thickening was more present in vape alone while neutrophilia is more present in LPS alone, and with vape + LPS resulting in worsening of both features. In the presence of 2-APB, this alveolar thickening and neutrophilia was diminished (Figure 3D). This also results in a significant diminishment of lung injury in the presence of 2-APB demonstrated by gross histological comparison and lung injury score. These data suggest that 2-APB has a prophylactic effect on vaping-dependent pulmonary disease and could potentially be used as a therapy.

### 2.4. Male Mouse LPS Exposure Data Demonstrate Differences in Cellularity and Inflammation Compared to Female Data

To investigate if disease burden is similar in males as in females, we vaped age-matched male mice for two weeks and then exposed them to LPS for 24 h. Post-infection animals (24 h) were sacked, and BALF and lungs were collected. Our flow cytometry analysis demonstrated a significant increase in neutrophil numbers observed in the vape + LPS group compared to the vape alone, which is similar to our female mice. However, this significance was not observed in LPS alone compared to vape + LPS (Figure 4A). Notably LPS alone and vape + LPS had higher neutrophil content and activation compared to the mock and vape, most likely due to the presence of infection. Significant differences in neutrophil activation were also observed between these two groups. Compared to females, the male mice appear to have slightly higher neutrophil numbers and activation (Figure 3). TNF-α and IL-6 levels were also investigated as described previously. A significant increase in IL-6 was observed in the vape + LPS group compared to vape alone group but not compared to LPS alone. However, no significant changes in TNF-α levels were detected (Figure 4B). The vape alone and LPS alone groups appeared to have less TNF- α activation while vape + LPS treatment resulted in more activation (though not statistically significant when compared to the mock group). When compared to the female data, it appeared that the male LPS treatment group had higher IL-6 levels while female mice displayed higher TNF-α levels. However, a similar result was not evident for the vape + LPS groups. Finally, H&E staining of lung sections demonstrated increased alveolar wall thickening and neutrophilia in the vape and vape + LPS groups (Figure 4C). The lung injury scoring data indicated a significant increase in damage with the vape alone lungs compared to the mock lungs. The males demonstrated no significant difference in the presence of LPS. However, when comparing the histopathology data, females appeared to have a worsened disease burden when vaped and challenged with only LPS (Figure 3D).

## 3. Discussion

E-cig use, or “vaping”, has gained widespread popularity and use among youth and young adults since ~2015 due to the prevalent marketing of JUUL and similar devices. As of 2023, approximately 2.8 million high-school- and middle-school-aged students have utilized nicotine-delivering tobacco products including E-cig products [12]. These data underline a need for research focused on understanding the full effects of E-cig use. Our previous research and research within this field have employed “vaped” e-liquids to investigate the negative effects of vaping. This research has shown that vaping is associated with inflammation, acute lung injury, and other pulmonary toxicities [16,28]. Though the findings of these research studies are important in understanding the effects of E-cig exposure, the biological relevance of these exposure models and routes has come into question as humans are not exposed to E-cigs as an e-liquid but rather as an e-liquid aerosol. Research has begun to focus more on the use of the aerosol or vapors produced from E-cigs as a more relevant exposure model as there may be chemicals and ROS that humans are only exposed to in aerosol form [17,30]. Here we employed the SCIREQ InExpose compact inhalation system (Figure 1B). Using this system, we were able to expose mice directly to the aerosol form of our selected E-cig of choice (JUUL “Menthol”) allowing a more biologically relevant model of “vaping”. The lung wet/dry weight ratio also demonstrated a significant increase in lung weight in the vape compared to the mock, further confirming the worsened disease prognosis (Figure 1C).

Through BCA analysis, we were also able to observe an increase in total protein present; however, the increase was not significant. Increases in lung wet/dry weight are a feature of lung damage typically accompanied by an increase in BALF protein [31,32]. The lack of significance present in total protein may be due to the length of the experiment. Previous research shows robust protein content in chronically E-cigarette-exposed mice over a time course of 3 to 6 months [33]; however, this increase was demonstrated in blood. This suggests that a longer time course may be needed for a large increase in protein to be observed. However, we demonstrated that lung injury can been seen in as little as 4 weeks through lung injury scoring and gross histological analysis in further experiments, suggesting the pathology associated with fluid accumulation and inflammatory cells. Additionally, blood may be a potent method of investigating total protein changes in addition to BALF in future experiments.

Utilizing this exposure model, we also wanted to investigate if the effects of vaping are present in as little as 2 weeks to create an acute exposure model (Figure 2A). For this exposure, we employed female mice, which produced similar data as in our previous experiment. There was a significant neutrophil increase in the vape + LPS group compared to vaping alone. However, the increase in neutrophil activation was not significant (Figure 2B). IL-6 and TNF-α were investigated due to being prevalent in pulmonary diseases and as a result of long-term vaping [26,30]. Significant increases in the vape + LPS group for both IL-6 and TNF-α corroborate our previous findings, indicating that vaping likely primes the lung for worsened inflammation/disease. These results were further corroborated by lung histology, illustrating increased neutrophilia and alveolar wall thickening in vape + LPS mice.

Having established a model for vaping-associated pulmonary disease exacerbation, we next questioned whether Ca^2+^ dysregulation was involved in this disease burden. We added the variable of including 2-APB, a known Ca^2+^ signaling inhibitor, to investigate the role Ca^2+^ signaling in vaping-dependent pulmonary disease. This is a relevant question to address as Ca^2+^ is an important mediator of many of biological processes, and studies have demonstrated differences in Ca^2+^ levels due to e-liquid exposure [28,34]. Also, our previous study utilizing JUUL “Menthol” suggested that exposure to this e-liquid results in increased intracellular Ca^2+^ levels [28]. This result suggests an influx (from an extracellular source) or efflux (from intracellular stores) of Ca^2+^ due to vaping. Because of this previous result, we sought to inhibit Ca^2+^ signaling to observe if inhibiting either the influx or efflux of Ca^2+^ would result in a better disease outcome. In the presence of 2-APB, we found a significant decrease in neutrophil number compared to vape + LPS (Figure 3B). Concomitantly, we observed an increase in macrophage numbers in the mice treated with 2-APB (Figure 3B). We found a significant increase in IL-6 in vape + LPS compared to vape only but no significant diminishment of this inflammatory burden in the presence of 2-APB (Figure 3C). With regards to TNF-α levels, there were no significant changes in expression, though 2-APB-treated animals trended towards an increase in TNF-α compared to LPS-treated animals. Histopathology analysis presented decreased lung thickening and neutrophilia after 2-APB treatment. This result matched our observed decrease in neutrophil number in the presence of 2-APB (Figure 3D). These data suggest that 2-APB may result in the diminishment of neutrophils along with a resulting increase in macrophages. Macrophages are responsible for releasing inflammatory markers such as IL-6, and previous research has demonstrated that exposing macrophages to condensed e-liquids results in an increased production of IL-6 and TNF-α [25,35]. These results indicate that 2-APB may be useful in mitigating the damage caused by vaping but not necessarily inflammatory cytokine activation.

However, our study has some admitted limitations, for example, our study surrounding 2-APB, which is a compound lacking specificity. This is due to 2-APB targeting multiple components of the Ca^2+^ pathway. 2-APB is a known inhibitor of intracellular inositol 1,4,5 trisphosphate (IP_3_) receptors, which play an important role in SOCE or the release of Ca^2+^ from intracellular stores [36,37]. 2-APB also has been shown to block or activate TRP channels and activate Orai-mediated Ca^2+^ entry [37,38]. These data underline a need to genetically analyze 2-APB targets in vitro using cell lines to better understand their effects. To this end in the future, we will employ macrophages and neutrophils utilizing further Ca^2+^ probing experiments in order to investigate 2-APB and other drugs as a potential treatment for vaping-dependent pulmonary inflammation. A second limitation of our study is the use of LPS. Though LPS is present on the surface of Gram-negative bacteria and in our study triggered a response, our use of LPS does not accurately represent the human route of exposure. A future direction to remedy this limitation will focus on the use of a Gram-negative bacterium such as *Klebsiella pneumoniae* as this is a common agent of human infection [39]. Additionally, in the future we will also include male and female mice while further studying the differences between their disease outcomes in the presence of 2-APB.

In summary, these data indicate that macrophages may play a beneficial role in the inflammatory response contributed by vaping. One theory suggests that Ca^2+^ regulation is responsible for macrophage polarization, and is therefore responsible for vaping-dependent pulmonary inflammation. Macrophage polarization leads not only to the mobilization, release, and phagocytic activity of macrophages, but may also trigger the shift of macrophages from the M0 state to the pro-inflammatory M1 state [40,41].

In conclusion, our model further confirms our previous research that vaping exacerbates pulmonary disease in mice. The exact mechanism is not known but is believed to be through Ca^2+^ signaling transduction. Vaping may cause a disruption of Ca^2+^ signaling, but through the inhibition of these pathway(s) we observe a diminishing effect on the acute lung injury caused by vaping and/or an infection. Our data demonstrate that females have worsened disease outcomes due to vaping compared to males, suggesting that diminishment in Ca^2+^ signaling dysregulation may result in this decrease in disease burden. This underlines a need to understand gender differences in vaping-dependent pulmonary diseases for ideal therapeutic treatment. Thus, our mouse model also presents a more biologically relevant model of inhalation with uses towards developing potentially therapeutic interventions for vaping-dependent pulmonary diseases.

## 4. Materials and Methods

### 4.1. Mice and Treatment

All mice were obtained from Jackson Laboratories (Bar Harbor, ME, USA). Young adult C57-BL/6J mice (6- to 8-week-old male and female) were used for all experiments. After being received, the mice were allowed to acclimate and recover from shipping stress for at least 1 week in the NCCU Animal Resource Complex, which is accredited by the American Association for Accreditation of Laboratory Animal Care. All animal care and use were conducted in accordance with the Guide for the Care and Use of the Laboratory Animals (National Institutes of Health), and mice were maintained at 25 °C and 15% relative humidity with alternating 12 h light/dark periods.

Once acclimated, mice were fitted into the SCIREQ InExpose system (SCIREQ Respiratory Equipment, Montreal, QC, Canada) and vaped using “Menthol” JUUL pods (JUUL, Washington, DC, USA). Manufacturer’s label information stated ingredients include only vegetable glycerin (VG), propylene glycol (PG), nicotine, flavoring, and benzoic acid, with each pod containing 0.7 mL of the flavored fluid at 3% nicotine. During the two weeks of exposure, appropriate mice were given 10 µL of a 500 µM stock of 2-APB (a calcium channel inhibitor, Sigma-Aldrich St. Louis, MO, USA). Following these treatments, mice were anesthetized via an i.p. injection of ketamine (100 mg/kg) and xylazine (50 mg/kg), and 20 µL of 0.4 mg/mL lipopolysaccharide (LPS) was delivered dropwise intranasally (IN) to mimic an infection (Sigma-Aldrich St. Louis, MO, USA). Mice were sacrificed after 24 h, and bronchoalveolar lavage fluid (BALF) and total lung were collected.

### 4.2. Bicinchoninic Acid (BCA) Protein Assay

At experimental endpoints, mice were sacrificed, BALF was collected, supernatants were isolated and total protein was analyzed. BALF total protein was evaluated using Pierce™ BCA Protein Assay Kit purchased from Thermo Fisher Scientific (Waltham, MA, USA).

### 4.3. Flow Cytometry

At experimental endpoints, mice were sacrificed, and BALF was collected. Cells were isolated for flow cytometry and extracellularly stained as either macrophages or neutrophils and for their activation. Antibodies (Siglec-F, Ly6G, F4/80 and CD11b) and Pacific Blue for viability were purchased from Thermo Fisher Scientific (Waltham, MA, USA).

### 4.4. Cytokine Analysis

At experimental endpoints, mice were sacrificed, BALF was collected, and supernatants were isolated for cytokine analysis. Inflammatory cytokine proteins were evaluated using ELISAanalysis (OptEIA, BD Pharmingen, Franklin Lakes, NJ, USA) and Graphpad Prism 9.

### 4.5. Histopathology

At the time points of experimental completion, mice were sacrificed. Lungs were then inflated with 1 mL of 10% neutral buffered formalin, removed, and suspended in 10% formalin for 12 h. Lungs were washed once in PBS and then immersed in 70% ethanol. Tissues were then embedded in paraffin, and three 5 μm sections (200 μm apart) per lung were stained with hematoxylin/eosin (H&E). Sections were evaluated blindly for gross pathology at 40× and 100× total magnification. Acute lung injury scoring was completed at 400× using the procedure outlined in the following paper [42]. Briefly, at least 20 pictures were taken of each lung at 400× total magnification. To generate the acute lung injury score, pictures were scored using 5 independent variables weighted according to relevance. These variables were as follows: neutrophils in the alveolar space, neutrophils in the interstitial space, hyaline membranes, proteinaceous debris filling the airspaces, and alveolar septal thickening. These variables were then summed and normalized by the number of fields to create a lung injury score between zero and one.

### 4.6. Statistics

All statistics were performed using GraphPad Prism 9 (La Jolla, CA, USA). All experimental samples were run in at least triplicate and statistical significance was calculated using unpaired *t*-tests with Welch’s correction.

## Figures and Tables

**Figure 1 ijms-25-01785-f001:**
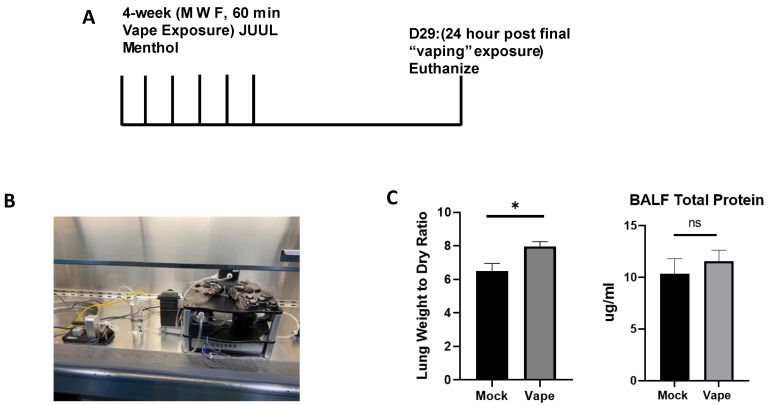
Establishing an acute vaping mouse model using the SCIREQ InExpose system. (**A**) Mouse model for acute vaping using the SCIREQ InExpose system (**B**). SCIREQ InExpose compact inhalation system in use. (**C**) Lung wet/dry ratio demonstrated significant changes in ratio of vaped when compared to mock groups. BALF total protein did not show a significant difference (* *p* < 0.05, ns = not significant).

**Figure 2 ijms-25-01785-f002:**
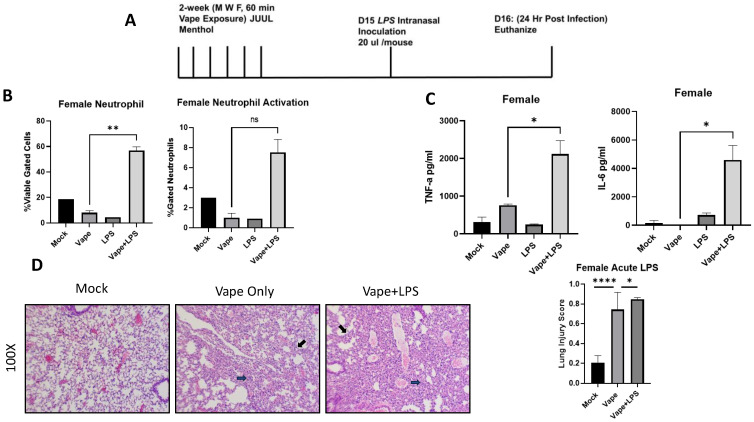
Acute vaping in vivo model shows vaping exacerbates lung cellularity and inflammation in the presence of infection. (**A**) Mouse model for acute vaping using the SCIREQ InExpose system. (**B**) Bronchoalveolar lavage fluid (BALF) was collected from mice, and cells and supernatant were separated by centrifugation. Cells were stained using antibodies for neutrophils and macrophages as well as their activation. The LPS group has lower neutrophil content and activation when compared to mock and vape alone groups. Vape + LPS groups have a higher neutrophil content and activation, indicating vaping plus infection (LPS) leads to increased cell response (** *p* < 0.01. ns = not significant). (**C**) BALF supernatant was evaluated for the abundance of TNF-α and IL-6 using ELISA. Female mice have higher levels of IL-6 and TNF-α. Vaping also appears to have an increased effect on IL-6 and TNF-α in the presence of LPS. Symbols and bars represent the mean ± SEM compared to the mock infected group (* *p* < 0.05, **** *p* < 0.001). (**D**) H&E staining of lung sections demonstrate alveolar wall thickening and increased neutrophilia in vape and vape + LPS groups when compared to the mock group. Alveolar wall thickening is indicated by black arrows while neutrophils are indicated by blue arrows. Lung damage was quantified using a lung injury scoring method. Significant increases in vape alone vs. mock as well as vape + LPS vs. vape are evident (mag = 100×).

**Figure 3 ijms-25-01785-f003:**
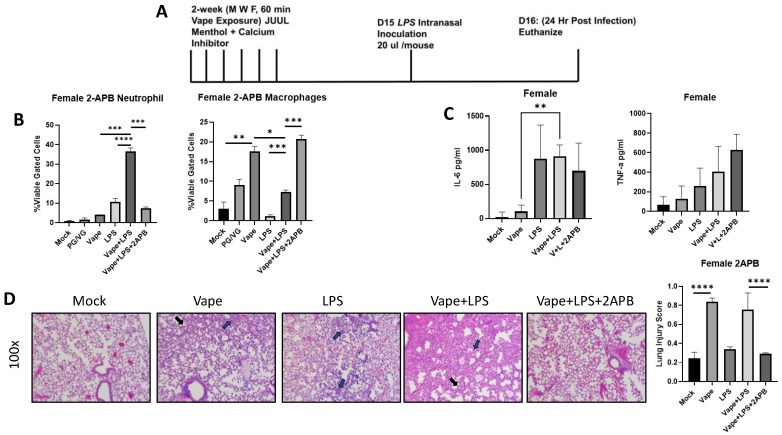
2-APB treatment mitigates alveolar damage caused by vaping or infections. (**A**) Mouse Model for Acute vaping using SCIREQ InExpose system. (**B**) Bronchoalveolar Lavage Fluid (BALF) was collected from mice and cells and supernatant were separated by centrifugation. Cells were stained using antibodies for neutrophils and macrophages as well as their activation. A significant increase in neutrophil migration into the alveolar space in the vape + LPS mice when compared to vape alone mice was observed; however, when observing the activation of these neutrophils, the increase was not significant (* *p* < 0.05, ** *p* < 0.01, *** *p* < 0.005, **** *p* < 0.001). (**C**) The inflammatory cytokines tumor necrosis factor alpha (TNF-a) and Interlukin-6 (IL-6) were then investigated using clarified BALF. When comparing the vape and vape + LPS groups, both cytokine levels were significantly increased. (**D**) Histopathology of lung tissues. H&E staining of sections of lung tissue isolated from mock, vape alone, LPS alone, vape + LPS, and vape + LPS + 2-APB. Alveolar wall thickening is present in vape and vape + LPS groups compared to mock. LPS and vape + LPS demonstrate neutrophilia compared to mock with vape + LPS having more visible lung injury. This alveolar thickening appears to be diminished in the vape + LPS + 2-APB group. Alveolar wall thickening indicated by black arrow and neutrophils indicated by blue arrow. Lung injury was further quantified through scoring. The data demonstrate that the increased in lung injury through vape alone and vape + LPS can be diminished by the presence of 2-APB (mag = 100×).

**Figure 4 ijms-25-01785-f004:**
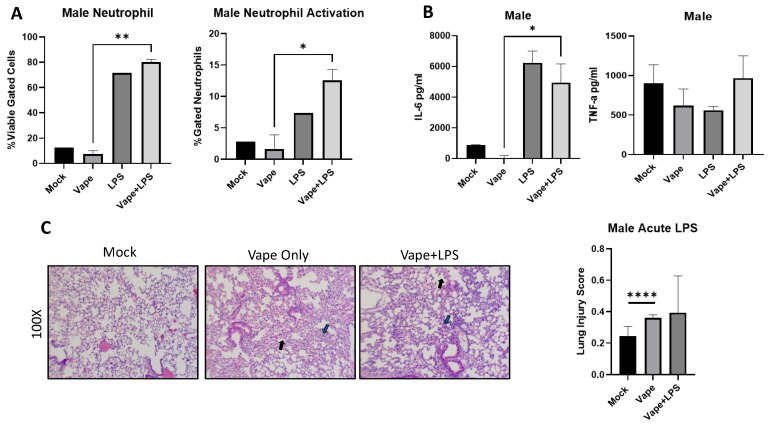
Male mouse LPS exposure data demonstrate differences in cellularity and inflammation compared to female data. (**A**). Bronchial alveolar lavage fluid (BALF) was collected from mice, and cells and supernatant were separated by centrifugation. Stained cells from flow cytometry demonstrated a significant increase in neutrophil numbers observed in the vape + LPS versus vape only. (**B**) The inflammatory cytokines TNF-α and IL-6 were then investigated using clarified BALF. A significant increase in IL-6 was measured in the vape + LPS group compared to vape only group. However, no significant changes in TNF-α levels were detected. (**C**). Histopathological staining of lung sections demonstrated increased alveolar wall thickening and neutrophilia in the vape and vape + LPS groups, with the mock and vape having significant differences based on lung injury scoring. Alveolar wall thickening indicated by black arrow and neutrophils indicated by blue arrow (mag+100×, * *p* < 0.05, ** *p* < 0.01, **** *p* < 0.001).

**Table 1 ijms-25-01785-t001:** Extracellular Staining Antibodies.

Antibody	Detection Channel (Excitation)	Use
SiglecF	FITC (488 nm)	Alveolar Macrophage
Ly6G	PE (488 nm)	Neutrophil
F4/80	APC-AF750 (638)	All Macrophages
CD11b	APC (638 nm)	Activation Marker
Live/Dead blue	PB450 (405 nm)	Cell Viability

## Data Availability

The data presented in this study are available on request from the corresponding author.

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
