# Peer review of "Vaping-Dependent Pulmonary Inflammation Is Ca2+ Mediated and Potentially Sex Specific"

_ijms, 2024, doi:10.3390/ijms25031785_

Round 1

Reviewer 1 Report

Comments and Suggestions for Authors

Major:

The manuscript by Shipman et al. reports a well-executed study of the effects of vaping Juul e-cigs on lung inflammation and injury in the mouse model. The authors show inhibition of vaping lung injury by the Ca++ signalling modulator 2-ABP, and an interesting effect of female gender on the severity of the injury. The work was design well and the data are sound, interesting and comprise an important contribution to the literature. 

The only possible deficiency in the paper note by this reviewer is lack of discussion of whether Ca++ signalling modulators such 2-ABP have potential for use in humans, possible off-target effects and "drugability" issues. Beyond that, critiques are minor.

Minor:

Figs 1-4: The Figures and data are otherwise clear and solid, but the Axis Labels are quite small font and difficult to read in current form. Its not clear if this will be better in journal format, so enlargment of the axis fonts would make reading easier.

Author Response

Dear Reviewer,  we addressed both reviewers comments in a consolidated document, and included edits in track changes upon the revised draft.

Reviewer 2 Report

Comments and Suggestions for Authors

Journal IJMS (ISSN 1422-0067)

Manuscript ID: ijms-2802116

Title: Vaping-dependent pulmonary inflammation is Ca2+ mediated and potentially sex specific

Authors: Jeffrey Giovanni Shipman , Rob Onyenwoke , Vijay Sivaraman *

Brief summary:

This study investigates the role of iCa signaling in LPS-induced ALI in vaping-pre-exposed mice, and potential sex differences in this disease process. The authors conclude that pre-exposure to vaping accelerates the subsequent development of LPS-induced ALI in both sexes, but with worse levels of inflammatory markers in females than males. Using the pharmacological inhibitor Ca signaling inhibitor 2-ABP, the authors identified a role for dysregulation in iCa signaling in this disease process.

Strengths:

-pre-exposure to vaping followed by LPS infection creates a clinically-relevant scenario

-use of an in vivo model

-addressing sex differences in clinical outcomes

-use of a nose-only inhalation model

Weaknesses:

--The main concerns revolve around inaccuracies of reporting the presented data, and some lack of rigor in dissecting the Ca response, which is stated as a major goal for this study.

--I am confused by Fig.1: on one hand it shows 4 weeks of vaping exposure, on the other it shows mouse take-down at day 15? Why does the X axis arrow continue past take-down of mice (also true for Fig. 2)? What exposure does “post final exposure” refer to? LPS or vaping?

The increased wet-dry lung ratio in Fig.1 suggest breakdown of barrier function. Were BAL total protein levels (or IgM levels, or any other molecular marker of barrier dysfunction) altered after vaping to support barrier breakdown?

--Fig. 2: An increase in BAL total cells, especially neutrophils, after 24 hrs and with the LPS dose used in this study, has been routinely observed by many studies. Why is there a decrease in BAL cells/neutrophils after LPS observed in this model? Similarly, why is there a decrease in BAL neutrophils seen after vaping exposure? Is the increase in TNF-a after vaping significant? Is the increase in IL-6 after LPS significant? Please comment on these findings in the Results as they appear significant based on the graphs. Please quantify the histology data (Fig. 2) with the lung injury score proposed in the Methods section. Why is the mock response for neutrophils in Fig. 3B almost 10-fold lower than for the same type of experiment in Fig. 2B? Why is the IL-6 response to vape+LPS about 5000pg/ml in Fig 2C but only <1000 pg/ml in Fig. 3C? Why is there an increase in TNF-a after vape+LPS in Fig 2C but not for the same condition in Fig. 3C?

--I am not sure how to integrate the 4-weeks vaping model in Fig. 1 with the 2-week vaping (+/- LPS) model in Fig. 2? They appear to me like unrelated models. What is the relevance of the 4-week model for the rest of the paper?

-- using the drug 2-APB as the sole approach to investigate Ca signaling is of rather limited value. 2-APB targets multiple Ca-dependent pathways, both intracellular (IP3, mitochondrial etc.) as well as extracellular (SOC, voltage gated Ca channels, TRP channels etc.), in addition to creating non-specific Ca leaks from intracellular stores, with additional significant differences depending on drug concentration, cell type, and species (e.g. PMID: 12153982 among many other papers). None of these topics are even discussed. The impact of the paper could be significantly improved by at least discriminating between intra- and extracellular Ca dependent mechanisms using more targeted pharmacology, genetic approaches, either in vivo or in vitro on cell types of interest (e.g. neutrophils, macrophages as pointed out in the manuscript).

--In Fig. 4 and lines 238-239, why is it a “surprise” to find increased neutrophils in males in vape+LPS compared to vape alone, when the same was observed in females? The major difference between males and females is in the LPS response.

--Lines 250-251 state that males have higher IL-6 levels than females, which is not true for the vape+LPS group (only the LPS group). It is also stated that males have higher TNF-a levels than females, which is not true for the vape+LPS group.

--given the reported differences in the inflammatory response between males and females, why is the role of iCa only investigated in female mice but not in males?

--drug concentrations (LPS, 2-ABP) should be reported directly, not as volumes of stock solutions.

--in addition to providing a reference, the components of the lung injury score should be listed in the manuscript.

-LPS does not truly reflect a clinically-relevant bacterial pneumonia model (compared to live bacteria) as no patient actually inhales LPS.

--how was 2-ABP administered?

Author Response

Dear Reviewer,  we addressed all reviewer comments in a consolidated response document, as well with track changes in the revised draft.

Round 2

Reviewer 2 Report

Comments and Suggestions for Authors

Thank for your response and the additional data/information.

It's interesting that lung wet:dry ratio changes, while the lack of BALF protein changes does not support a breakdown in barrier function. Can the authors add an explanation of what may cause this discrepancy to the Discussion? 

Author Response

We address these comments within the revised draft in lines 314-325.

Other published work has observed differences in total protein concentration in lungs due to time course of chronic vaping, and these time courses are longer than our experiments. We believe that the difference between significant lung weight changes and observed increase in inflammatory cellularity and no appreciable total protein difference could be based on inflammatory fluid accumulation and exudate inflammatory cells, that are not appreciated in total protein assessed by BCA assay.